# Comparative Analysis of the Physicochemical and Biological Characteristics of Freeze-Dried PEGylated Cationic Solid Lipid Nanoparticles

**DOI:** 10.3390/ph16111583

**Published:** 2023-11-09

**Authors:** David A. Narváez-Narváez, María Duarte-Ruiz, Sandra Jiménez-Lozano, Cristina Moreno-Castro, Ronny Vargas, Anna Nardi-Ricart, Encarna García-Montoya, Pilar Pérez-Lozano, Josep Mª Suñé-Negre, Cristina Hernández-Munain, Carlos Suñé, Marc Suñé-Pou

**Affiliations:** 1Department of Pharmacy and Pharmaceutical Technology, and Physical Chemistry, Faculty of Pharmacy, University of Barcelona, 08028 Barcelona, Spain; dnarvana20@alumnes.ub.edu (D.A.N.-N.); rvargamo20@alumnes.ub.edu (R.V.); annanardi@ub.edu (A.N.-R.); encarnagarcia@ub.edu (E.G.-M.); perezlo@ub.edu (P.P.-L.); jmsune@ub.edu (J.M.S.-N.); marcsune@ub.edu (M.S.-P.); 2Department of Molecular Biology, Institute of Parasitology and Biomedicine “López-Neyra” (IPBLN-CSIC), 18016 Granada, Spain; maduduru@ipb.csic.es (M.D.-R.); sandra_11406@ipb.csic.es (S.J.-L.); cristina.moreno.castro@ulb.be (C.M.-C.); 3ULB Center for Diabetes Research, Faculty of Medicine, Université Libre de Bruxelles, 1050 Brussels, Belgium; 4Department of Pharmaceutical Technology, Faculty of Pharmacy, University of Costa Rica, San José 11801, Costa Rica; 5Pharmacotherapy, Pharmacogenetics and Pharmaceutical Technology Research Group, Bellvitge Biomedical Research Institute (IDIBELL), 08908 Barcelona, Spain; 6Department of Cell Biology and Immunology, Institute of Parasitology and Biomedicine “López-Neyra” (IPBLN-CSIC), 18016 Granada, Spain; chmunain@ipb.csic.es

**Keywords:** cationic solid lipid nanoparticles, cSLNs, poly(ethylene glycol), PEG, freeze-drying, lyophilization, SLNplexes, gene therapy, stability, morphology

## Abstract

Cationic solid-lipid nanoparticles (cSLNs) have become a promising tool for gene and RNA therapies. PEGylation (PEG) is crucial in enhancing particle stability and protection. We evaluated the impact of PEG on the physicochemical and biological characteristics of cholesteryl-oleate cSLNs (CO-cSLNs). Several parameters were analyzed, including the particle size, polydispersity index, zeta potential, shape, stability, cytotoxicity, and loading efficiency. Five different formulations with specific PEGs were developed and compared in both suspended and freeze-dried states. Small, homogeneous, and cationic suspended nanoparticles were obtained, with the Gelucire 50/13 (PEG-32 hydrogenated palm glycerides; Gelucire) and DSPE-mPEG2000 (1,2-distearoyl-phosphatidylethanolamine-methyl-polyethyleneglycol conjungate-2000; DSPE) formulations exhibiting the smallest particle size (~170 nm). Monodisperse populations of freeze-dried nanoparticles were also achieved, with particle sizes ranging from 200 to 300 nm and Z potential values of 30–35 mV. Notably, Gelucire again produced the smallest particle size (211.1 ± 22.4), while the DSPE and Myrj S100 (polyoxyethylene (100) stearate; PEG-100 Stearate) formulations had similar particle sizes to CO-cSLNs (~235 nm). The obtained PEGylated nanoparticles showed suitable properties: they were nontoxic, had acceptable morphology, were capable of forming SLNplexes, and were stable in both suspended and lyophilized states. These PEG-cSLNs are a potential resource for in vivo assays and have the advantage of employing cost-effective PEGs. Optimizing the lyophilization process and standardizing parameters are also recommended to maintain nanoparticle integrity.

## 1. Introduction

In the last 30 years, nanotechnology has become a novel interdisciplinary scientific field that has promoted the development of several nanostructures with the potential to provide controlled drug release and targeted delivery of active agents [1]. Along with the development of new and efficient technologies, nanoparticles (NPs) have attracted great interest and have become a promising tool to establish new therapeutic routes for clinical use, such as gene and ribonucleic acid (RNA) therapies [2,3].

Currently, there are two types of delivery systems, viral and non-viral vectors, which each have specific advantages and limitations [4,5]. Although viral vectors offer high transfection efficacy, they also present several disadvantages, such as marked immunogenicity, insertional mutagenesis [6], deoxyribonucleic acid (DNA) package size limitations, and non-specific effects [7,8]. On the other hand, non-viral vectors (physical or chemical) do not present these problems and are safer alternatives for gene therapy applications [9]. Furthermore, in the last few years, non-viral vectors have shown comparable efficacy to viral vectors. A notable example of this was observed with coronavirus disease 2019 (COVID-19) vaccines, where lipid nanoparticle-based formulations presented superior efficacy compared to conventional viral vector-based vaccines [10].

In this context, solid lipid nanoparticles (SLNs) have increased in importance in recent years and are among the most promising nanoparticle-based methods for nucleic acid delivery [1,8,11]. Cationic solid lipid nanoparticles (cSLNs) are biodegradable and biocompatible non-viral lipid-based nanoparticles with a positive surface charge, that are capable of forming complexes with DNA/RNA (SLNplexes) [4,12]. However, barriers such as complex and difficult-to-scale manufacturing processes, high cost, and low transfection efficiency continue to hamper the widespread application of nanotechnology for clinical purposes in humans, particularly when compared to viral vectors [13]. Nonetheless, increasing efforts have been undertaken to improve these issues [14,15,16]. In fact, lipid-based nanoparticle formulations are already on the market for treating human diseases [17].

Surface modification is a commonly employed strategy to improve the bioavailability of particles. In this context, the addition of poly(ethylene glycol) (PEG) to the main formulation offers numerous advantages since it is a synthetic, biocompatible polymer extensively utilized in the pharmaceutical, cosmetic, and food industries [18]. Currently, PEG is the most commonly used polymer for coating nanoparticles, and it might be considered the gold standard of stealth polymers [19]. Most commercially available polymer-based stealth drug-delivery systems and those in advanced clinical trials contain PEG, and no other synthetic polymer has yet reached this status [19]. For instance, recently approved lipid nanoparticle drug products for human medical use (onpattro-patisiran-SLN and mRNA-lipid nanoparticle (LNP) COVID-19 vaccines) are PEGylated [20].

PEG, as a surface-modifying agent, “masks” nanoparticles, creating a steric barrier in the form of a PEG layer, which prevents particle aggregation and shields the nanoparticles from interactions with proteins and biomacromolecules (such as nucleases) in the blood, cytosol, and lysosomes [7]. The repulsive properties of PEG play an important role in mitigating enzymatic degradation and preserving the structural integrity of PEG-SLNplexes [19,21]. PEGylation also prevents the adsorption of opsonins (antibodies) onto the particles, resulting in increased stealth efficiency, reduced immunogenicity, prolonged circulation time, and the prevention of aggregation during circulation [18,22,23].

Starting from established formulations of cSLNs with cholesteryl-oleate (CO-cSLNs), which are manufactured without the use of an organic solvent and have demonstrated their efficacy and safety in vitro [24,25], this study was designed to improve the current formulation for the possible administration of RNA/DNA in vivo. The incorporation of polyethylene glycol (PEG) in the nanostructures may improve their physicochemical, pharmacokinetic, and/or transfection properties due to its amphiphilic nature, which promotes the transformation of nanoparticles into hydrophilic particles, thereby improving their blood circulation time. These improvements increase the likelihood of a specific drug reaching its target site before being identified as foreign and eliminated from the body [19,22].

Therefore, the overall goal of this study was to evaluate the influence of incorporating a new excipient (PEG) on the stability, physicochemical properties, morphology, and biological characteristics of CO-cSLNs, both before and after lyophilization. Since the incorporation of PEG substantially increases manufacturing costs, five distinct PEG excipients were tested to explore feasible formulations that utilize cost-effective PEGs, potentially widening their commercial accessibility. This comprehensive assessment compared PEGylated and non-PEGylated cSLNs, contributing to a deeper understanding of their performance and potential applications.

## 2. Results and Discussion

### 2.1. Physicochemical Properties of the Formulations

#### 2.1.1. Factorial Study

The factorial study aimed to determine the optimal quantity of PEG to be incorporated into the lipid matrix, revealing variations in the physicochemical characteristics, particularly the particle size, based on the amount of PEG added. Among the tested formulations, two (Gelucire and Myrj 52) produced the smallest particle sizes (~225 nm) with a PEG quantity of 150 mg (Table 1). Moreover, the Myrj S100 formulation produced the smallest particle size (222.8 ± 10.1 nm) when 250 mg of PEG was utilized. However, when employing 150 mg, no notable variations in particle size were observed (227.6 ± 7.7 nm). Furthermore, the formulation containing Brij S100 produced the smallest particle size (166.2 ± 23.1 nm), accompanied by the highest employed quantity of PEG (350 mg; Table 1).

The incorporation of polymers, such as PEG, onto the nanoparticle surface can enhance colloidal stability and prolong circulation time through steric stabilization [22,26]. These effects are influenced by the molecular weight and structure characteristics of PEG and are directly correlated with the length and surface density of the PEG employed [22]. Notably, larger PEG chains could enhance the colloidal stability of the SLNs, preventing particle agglomeration and subsequently resulting in a reduction in particle size [18]. In this sense, the number of ethylene glycol repetitions in each PEG, which varies across different formulations, can substantially impact the physicochemical characteristics of the nanoparticles. Formulations containing Gelucire and Myrj 52 have fewer ethylene glycol repetitions (~30–40 times) than Brij S100 and Myrj S100, where the number of repetitions reaches 100. In the case of Brij S100, as a greater quantity of PEG is used, particle sizes tend to decrease. The extensive PEG coverage on the nanoparticle surface probably results in a more pronounced steric effect between nanoparticles.

However, it is imperative to exercise caution regarding the appropriate quantity of PEG to be incorporated. Excessive amounts may compromise cell transfection efficacy and reduce circulation time due to decreased PEG mobility and flexibility [22,23]. Hence, we selected 150 mg of PEG to integrate into the final formulation (SA, 140 mg + CO, 210 mg + PEG, 150 mg), since we considered this to be the optimal amount for efficient PEGylation of the nanoparticles. The four developed formulations exhibited satisfactory physicochemical properties, resulting in populations of monodisperse nanoparticles (PdI < 0.2) with a cationic surface charge (ZP between 25 and 40 mV) suitable for the formation of SLNplexes [8] (Table 1). PEGylation’s effect is further discussed in Section 2.2.

#### 2.1.2. Particle Size, Polydispersity Index and Zeta Potential

We successfully obtained a substantial population of small and homogeneous nanoparticles in all formulations. Nonetheless, the particle size was higher than our initial expectations (Table 2, original protocol). It has been reported that the particle size could be effectively reduced by increasing both the duration (min) and the stirring speed (rpm) during the manufacturing process. This size reduction might be attributed to several factors, including the reduction in the residence time of reactants in the system and the application of shear forces which disperse and fragment larger particles into smaller ones, promoting homogeneous nucleation and enhanced mixing that lead to a better dispersion of the active ingredients, allowing nanoparticles to form uniformly throughout the solution [27,28,29].

Thus, the stirring time and speed were raised from 10 to 15 min and from 20,000 to 24,000 rpm, respectively. Consequently, we achieved a particle size reduction ranging from 30 to 50 nm compared to the original values, with the formulations with Gelucire and DSPE having the smallest particle sizes of ~170 nm, followed by both Myrj formulations (52 and S100), with particle sizes of ~180 nm, and Brij S100, with a particle size of 190 nm. Even the CO-cSLNs exhibited a reduction in particle size, from 237.4 ± 4.9 nm to 202.1 ± 10.7 nm (Table 2).

In addition, in all formulations, the PdI values were less than 0.2 (Table 2, modified protocol), indicating a single population of PEG-cSLNs (monodisperse population) without aggregates. Concerning zeta potential, all PEG-cSLNs showed a positive surface area, with Myrj 52 having the highest ZP (31.4 ± 4.4 mV) and Brij S100 having the lowest ZP (25.1 ± 2.4 mV). The other three formulations (DPSE, Gelucire and Myrj S100) presented ZP values of ~27 mV (Table 2, modified protocol). Each PEG determined unique physicochemical characteristics for each formulation. The different polymers with different conformations produced differences in the nanoparticle surfaces, affecting the size, dispersion, and charge [23].

### 2.2. Impact of PEG Incorporation in the Formulation

Statistical analysis showed that the PEGylation of CO-SLNs affected the physicochemical properties of the nanoparticles. The particle size was reduced significantly in all PEG-cSLNs except in the formulation with Brij S100 (Figure 1A). No significant differences in the PdI were found, except in the formulation with Brij S100 (Figure 1B). Last, the surface charge decreased significantly due to PEGylation in all five developed formulations (Figure 1C).

The incorporation of a new excipient into an already established formulation to develop CO-cSLNs is a challenge since it affects the physicochemical properties of the nanoparticles. The addition of PEG to the original formulation affected particle size owing to the steric hindrance effect that the polymer performs [18,30]. This effect makes PEG-cSLN smaller (~180 nm) than CO-cSLN (~200 nm) by preventing agglomeration. The incorporation of PEG onto the surface also affected the surface charge of the nanoparticles, possibly due to surface area reduction, which decreases the charge [18].

In this context, statistical analysis showed that PEG-cSLNs developed with the modified protocol, including CO-SLNs, exhibited a significant reduction in particle size (Figure 2A). The PdI remained stable for the two protocols used, and no significant differences were found (Figure 2B). Only the formulations with Gelucire and CO presented significant differences in surface charge, while the other PEGs did not present differences (Figure 2C).

Particle size is a critical attribute that is directly related to the PdI: as the dispersion increases, the agglomeration and consequently the size of nanoparticles increase. These two parameters indicate the quality of the particles with respect to the size distribution, which is the main physicochemical attribute influencing endocytosis-dependent cellular uptake. The mechanisms of internalization depend on the nanoparticle size, surface properties (charge), shape, and types of cells involved [31,32].

### 2.3. Freeze-Drying of PEG-cSLNs

#### 2.3.1. Glass Transition Temperature (Tg)

The five formulations developed with 150 mg of PEG were lyophilized to facilitate transport and maintain physicochemical characteristics in the long term, improving their storage stability and shelf life. Prior to lyophilization, we determined the glass transition temperature (Tg) of each formulation to establish the correct parameters of the lyophilization process. For an optimal process, Tg should be below the collapse temperature (Tc) of the PEG-cSLNs. Tc is the maximum temperature at which the product can reach sublimation. Freeze-drying requires low temperatures and pressures to eliminate the solvent (in this case, water) in which the PEG-cSLNs are suspended through the sublimation process [33].

The lowest collapse temperature obtained among the formulations was −38 °C (Appendix A and Appendix A). Thus, for the entire product to be well lyophilized, the process should be performed at a minimum of 10−20 degrees lower than the aforementioned temperature [30]. Therefore, the freezing temperature used in the lyophilization cycle was −55 °C to ensure vitrification of the trehalose matrix and the appropriate lyophilization of all formulations [26].

#### 2.3.2. Lyophilization of the PEG-cSLNs

The physicochemical analyses showed the successful production of monodisperse populations of PEG-cSLNs (PdI ≤ 0.2) with particle sizes between 200 and 300 nm and Z potential values of 30–35 mV (Table 3). Among the formulations, Gelucire exhibited the smallest particle size (211.1 ± 22.4). Interestingly, the DSPE and Myrj S100 formulations demonstrated particle sizes comparable to those of CO-cSLNs (~235 nm). On the other hand, Myrj 52 displayed a higher particle size (280 ± 22.7 nm), while Brij S100 had the highest particle size overall (315.1 ± 15.1 nm), as presented in Table 3.

However, statistical analysis revealed a significant increase in the particle size of the freeze-dried PEG-cSLNs compared to the suspended PEG-cSLNs. Notably, even the CO-cSLNs exhibited a considerable size increase (Figure 3A). The stressful conditions of the freeze-drying process affected the physicochemical characteristics of all of the PEG formulations compared to CO-cSLNs. Previous studies also reported a rise in particle size after freeze-drying, providing further evidence of the effects of lyophilization on nanoparticles [26,30,33,34,35,36].

Regarding the PdI, no significant differences were observed between the suspended and freeze-dried PEG-cSLNs of the Gelucire formulation and the two Myrj formulations (52 and S100), suggesting the successful formation of monodisperse PEG-cSLNs without aggregates in these cases. Nevertheless, significant differences were observed for the other two PEGylated formulations (Brij S100 and DSPE), including CO-cSLNs (Figure 3B), revealing the need for further optimization of the freeze-drying process parameters, since steric stabilization might not guarantee the maintenance of physicochemical characteristics during the lyophilization process [37].

Concerning ZP, an increase was observed in all formulations except for the Myrj 52 formulation, where the potential remained stable (Table 3). This might be an advantage for PEGylated cSLNs due to the substantial impact of PEGylation on nanoparticle surface charge. Despite the increase in ZP, no significant differences were found for the Gelucire formulation and the two Myrj formulations (52 and S100) (Figure 3C). In contrast, significant increases in ZP were observed for the other PEGylated formulations (Brij S100 and DSPE) as well as the CO-cSLNs, indicating the influence of the freeze-drying process on ZP. This feature could be important in ensuring the formation of SLNplexes.

Freeze-drying is an alternative to improve the stability and feasibility of nanoparticles due to the poor stability of these colloidal systems. However, it is well documented that the freezing step is the most aggressive part of the process (crystal ice formation) and increases the risk of aggregation and/or coalescence, decreasing the stability of the nanoparticles [26,33,36]. To counteract this effect, lyoprotectants or cryoprotectants (amorphous disaccharides) are crucial because they protect and stabilize nanoparticles against freezing and drying stress during lyophilization, thus reducing aggregation, shortening reconstitution time and decreasing residual moisture [26,36]. These agents form a stable amorphous glassy matrix (vitrification hypothesis) that interacts with the surface of the nanoparticles, incorporating and immobilizing them and preventing their fusion or aggregation [30,38].

Nonetheless, the vitrification effect alone is insufficient to preserve the nanoparticles, as their physicochemical properties are also influenced by PEGylation, as we have previously demonstrated. It has been reported that a PEG-lipid with a longer PEG chain (MW 5000) and a shorter lipid scaffold (dimyristoyl glycerol) exhibited enhanced cryoprotective abilities (DMG-PEG5000) [36]. In this context, cSLNs with longer PEG chains (DSPE and Myrj S100) had particle sizes similar to those of CO-cSLNs, potentially due to the cryoprotective effects of PEG. It is essential to consider the characteristics of the PEG layer since they determine the stealth properties based on the flexibility of the PEG chains on the nanoparticle surface. The efficacy of a nanoparticle’s coating depends on the length and surface density of the PEG chains [22]. Furthermore, we observed that by optimizing the lyophilization process and standardizing the different parameters of the procedure, it is possible to maintain the nanoparticle’s physicochemical characteristics and integrity (unpublished data).

### 2.4. Morphology of Suspended and Freeze-Dried PEG-cSLNs

TEM images confirmed the presence of a significant proportion of spherical and well-defined PEG-cSLNs in both suspended and freeze-dried states. These images revealed the presence of SLNs with a solid matrix and the absence of amorphous structures in all developed formulations (Figure 4), indicating that the lyophilization process had no discernible impact on the nanoparticle shape. Moreover, the morphology was substantially different among the formulations, as the physicochemical characteristics were influenced by the specific PEGylated excipients employed in the nanoparticle matrix. The shape most widely used in designing drug delivery carriers is spheres. Nevertheless, particles with worm-like shapes, such as ellipsoidal, cylindrical, and discoidal shapes, or filomicelles, can achieve better accumulation within tumors [39]. In addition to particle size, PdI, and Z potential, shape is considered a crucial parameter, as it also determines the behavior of particles in various processes, including blood circulation, targeting, cellular uptake, and intracellular trafficking [22]. The interaction between shape and cell specificity is critical for cellular uptake, since shape also affects biological responses and enhances the targeting of antibody-coated nanoparticles to the endothelium [32].

### 2.5. Stability Study of the PEG-cSLNs

#### 2.5.1. One-Month Stability

Stability studies were carried out over 30 days at 4 °C and 25 °C to determine the temporal stability of suspended PEG-cSLNs. PEG-cSLNs in all formulations exhibited better stability at 4 °C than 25 °C since the SLN population remained monodisperse (PdI < 0.2) throughout the 30 days (Figure 5). However, the PdI values of all formulations at 25 °C and one (Brij S100) at 4 °C showed a correlation with the elapsed time (Appendix A). Thus, the PdIs increased over time, with indices exceeding 0.2 at approximately day 20.

Similarly, the particle sizes of all formulations were found to be smaller at 4 °C than at 25 °C. Nonetheless, it is notable that there was an increase in size compared to the initial manufacturing day (Figure 6). Consequently, a correlation was observed between the elapsed days and the particle sizes for all formulations at both temperatures. As the days passed, the size of the PEG-cSLNs progressively increased (Appendix A).

Finally, the ZP values of the PEG-cSLNs remained within the required range of 25–45 mV during the 30-day storage period (Figure 7). Moreover, correlations were observed between the ZP and the elapsed days at 4 °C for all formulations, except for the Brij S100 formulation (Appendix A). Thus, as the days passed, the ZP of the PEG-cSLNs increased. From approximately day 23, the Z potential increased.

Parameters such as particle size and size distribution (PdI) are important factors in evaluating the stability of aqueous colloidal systems [31]. In this context, Suñé-Pou et al. (2019) reported that CO-cSLNs began to aggregate from day 18, reaching maximum aggregation on day 24 at 4 °C. For the best performance, they recommended storing these nanoparticles for a maximum of 15 days at 4 °C. Nonetheless, PEG-cSLNs were more stable (~30 days) than the original formulations due to favorable factors such as the steric stability conferred by PEG and the surface charge established by the cationic lipid used.

Steric stabilization reduces the tendency of particles to aggregate, producing formulations with increased stability during production, storage, and application [19,23]. Moreover, surface charge might improve stabilization due to the repulsive electrostatic forces that exist between nanoparticles [30,40]. The cationic lipid contains amines whose nitrogen atoms become protonated when exposed to an aqueous medium, charging the nanoparticle surface [24]. Therefore, increasing the ZP provides greater stability by increasing electrostatic repulsion [40].

These two factors played a crucial role in enhancing the overall stability of these PEG-cSLNs [33]. Nevertheless, nanoparticles are more susceptible to changes in stability than microsystems due to their larger specific surface area [31,40]. The increased surface area makes nanoparticles more prone to aggregation, which becomes an important issue in comparison to other microscale drug delivery systems [31,40]. However, despite irreversible nanoparticle aggregation driven by van der Waals attractive forces [33], we were able to obtain satisfactory stability for these nanoparticles at 4 °C.

#### 2.5.2. One-Year Stability

In the same way, stability studies of freeze-dried PEG-cSLNs were performed at 4 °C and 25 °C for one year. Similar to suspended PEG-cSLNs, the stability of all PEG formulations was better preserved at 4 °C than at 25 °C since the PdI values remained at approximately 0.2 (Figure 8). These results indicate that the reconstituted PEG-cSLNs maintained a monodisperse population dispersion for one year, except for the formulation with Brij S100, which was the PEG most affected by time. Furthermore, a correlation was observed between PdI and the elapsed time for the formulations with Gelucire and Myrj 52 at 4 °C (Appendix A). Nonetheless, no correlation was found for the remaining formulations (Brij S100, DSPE, and Myrj S100). This factor might be attributed to the longer carbon chains present in these PEGs, particularly DSPE, which leads to an increased repulsion force owing to the steric effect on the completely coated cSLNs [18,41].

Similarly, the sizes of the PEG-cSLNs (230–250 nm) were stable at 4 °C in all formulations, including that with Brij S100 (Figure 9). Surprisingly, in some cases, the PEG-cSLNs were smaller at 25 °C than at 4 °C, while in others, they were larger (DSPE, Gelucire, Myrj 52 and S100), indicating instability at room temperature. This suggests that maintaining the nanoparticles at 4 °C is the most effective approach to prevent their agglomeration. Moreover, no correlation was found between particle size and elapsed time at either temperature analyzed, except in the formulation with Brij S100 at 4 °C (Appendix A). This means that the particle size remained stable throughout the entire year.

Interestingly, in some formulations, freeze-dried PEG-cSLNs exhibited a higher surface charge at 25 °C than at 4 °C (DSPE, Gelucire, Myrj 52 and S100; Figure 10). Nonetheless, all ZP values remained within the required range (25–45 mV) over the elapsed year in all formulations, as confirmed by subsequent biological tests assessing the electrostatic binding of genetic material to the nanoparticles. The purpose of preserving long-term stability and extending the shelf life of nanoparticles was achieved, although the freeze-drying process has not yet been fully standardized and optimized. However, through the improvement of process parameters, it is possible to maintain the integrity of the physicochemical characteristics of the nanoparticles and further enhance their long-term stability.

### 2.6. Biological Assays

#### 2.6.1. Cytotoxicity of PEG-cSLNs

The effects of the PEG-cSLNs on cellular viability/cytotoxicity were assessed. The results reveal that at a volume of 10 μL, all formulations in both states (suspended and freeze-dried PEG-cSLNs) demonstrated high cell viability, as evidenced by the absence of toxicity toward HEK293T cells in the IP test (Figure 11). In nearly all formulations tested at this volume, cell viability exceeded 90%. These outcomes suggest that under specific experimental conditions, these PEG-cSLNs do not induce harmful effects on human cells cultured in vitro. Conversely, higher toxicity was observed in practically all formulations when tested at a volume of 20 μL. Interestingly, the freeze-dried formulation with DSPE showed enhanced viability at a volume of 20 μL compared to 10 μL. Notably, Myrj S100 seems to be the most promising formulation, as both states exhibited cell viability over 90% at both tested volumes.

In vitro biological viability assays of CO-cSLNs have been reported previously [8,25]. To investigate the influence of PEGylation on their in vitro behavior using different PEG excipients, our study aimed to assess the cytotoxicity of these PEG-cSLNs in human cells under specific experimental conditions. The results indicate the absence of cytotoxic effects of these SLNs on human cells, providing a favorable indication to proceed with the formation of complexes with genetic material at a specific volume of nanoparticles.

#### 2.6.2. SLNplex Formation with PEG-cSLNs

To assess the nucleic acid binding efficiency of PEG-cSLNs, gel electrophoresis retardation assays were performed. The presence of unbound free DNA in the gels reflects the binding capacity of the PEGylated nanoparticles (loading efficiency). All PEG-cSLN samples (suspended and freeze-dried SLNs) formed SLNplexes when the amount of DNA was between 0.5 and 1.0 μg (Figure 12). In contrast, PEG-cSLNs were unable to form sufficient complexes with more than 2 μg of DNA, as a considerable amount of free DNA was detected in the gel. Nonetheless, the formulations with suspended SLNs formed complexes in all amounts except for the Myrj S100 formulation.

The capacity of CO-cSLNs to form DNA complexes has been reported previously [24,25]. Our results confirmed the suitability of these PEG-cSLNs for DNA binding, in both their suspended and freeze-dried states, despite the inclusion of PEGylation. While PEGylation reduced the surface charge (ZP), which may have affected the binding capacity, the remaining positive charge was enough to facilitate the formation of SLNplexes in all PEG formulations. Notably, the ZP increased after freeze-drying, improving the ease of SLNplex formation.

## 3. Materials and Methods

### 3.1. Materials

The following materials were used to synthesize the nanoparticles. The lipid matrix was composed of stearic acid (EMD Millipore, Billerica, MA, USA) and cholesteryl oleate (Tokyo Chemical Industry Co., Tokyo, Japan). Additionally, we selected five different PEGylated excipients: polyoxyethylene (100) stearyl ether- PEG100SE (Brij S100, Sigma-Aldrich Co., St Louis, MO, USA); PEG-32 hydrogenated palm glycerides (Gelucire 50/13, Gattefossé SAS, Saint-Priest, France); polyoxyethylene (40) stearate-PEG 40 monostearate (Myrj 52, Sigma-Aldrich Co); polyoxyethylene (100) stearate-PEG100S (Myrj S100, Croda Iberica S.A., Fogars de la Selva, Spain) and 1,2-distearoyl-phosphatidylethanolamine-methyl-polyethyleneglycol conjugate-2000 (DSPE-mPEG2000, Muse Chem, Fairfield, NJ, USA) for incorporation into the oil phase (dispersed phase) of each formulation after a bibliographic search [18,42,43,44] and an analysis of their molecular and solubility characteristics, hydrophilic-lipophilic balance, theoretical and experimental compatibility between the components, and price.

The hydrophilic components of the aqueous phase (continuous phase) were poloxamer 188 (Sigma-Aldrich Co.) as a surfactant (100 mg) and octadecylamine (Acros Organics, Geel, Belgium) as the cationic lipid (600 mg) used as a charged carrier. Finally, all the components were mixed with ultrapure water (EMD Millipore).

### 3.2. Method of Preparation of PEG-cSLNs

A factorial study involving different PEG quantities was conducted to determine the precise composition of the lipid core (stearic acid, SA + cholesteryl oleate, CO + PEG) based on their distinct physicochemical characteristics. The PEG-cSLNs were synthesized using an oil-in-water (o/w) emulsion technique based on the hot microemulsification method [45,46], with certain modifications that involved increasing the duration and the stirring speed within the manufacturing process to reduce particle size. Briefly, all the components were heated and melted at 10 °C above their melting point (~80 °C) in two separate beakers (5 mL and 50 mL) corresponding to the oil and water phases. Once the components of the lipid matrix were melted, they were poured and mixed into the aqueous phase solution (20 mL of ultrapure water + cationic lipid + surfactant) to create a hot emulsion. This mixture was high-speed stirred at 20,000 and 24,000 rpm with a digital ULTRA-TURRAX^®^ IKA^®^ T-25 equipped with a disperser S25N-8 G rotor Ø 8 mm (Staufen, Germany), for 10 and 15 min, respectively.

The hot emulsion obtained was rapidly dispersed into 125 mL of cold water (1–4 °C) under continuous high-speed stirring, employing the same conditions mentioned before, to generate the core solidification that forms the PEG-cSLNs. The resulting microemulsion was centrifuged at 19,000× *g* at 4 °C for 20 min (Digicen 20 R, Ortoalresa, Madrid, Spain) to remove any excess excipient. Finally, we double-filtered the nanoparticle suspension through 43–48 μm and 7–9 μm qualitative filter papers, respectively (FILTER-LAB^®^, Filtros ANOIA, S.A., Barcelona, Spain). The final SLN suspension (~80 mL) was dispensed into 5 mL glass vials (Vidrio Soplado Manuel Pérez, S.A., Barcelona, Spain) and stored at 4 °C until the freeze-drying process began.

### 3.3. Physicochemical Characterization of the PEG-cSLNs

#### 3.3.1. Size Characterization

The physicochemical properties of both the suspended nanoparticles in an aqueous medium and the reconstituted freeze-dried PEG-cSLNs with 4 mL of Milli-Q water were analyzed. Particle sizes and PdIs were determined by means of dynamic light scattering (DLS) with a Zetasizer Nano ZS90 (Malvern Panalytical Ltd., Malvern, UK) by examining the hydrodynamic particle diameter (z-average), which is expressed as the median volumetric particle diameter [47]. All samples were measured in triplicate to obtain the average nanoparticle size distribution and standard deviation with a mean value calculated in nanometers (nm).

Suspended formulations were characterized on the production day, whereas freeze-dried PEG-cSLNs were stored at 4 °C until further analysis. Measurements were performed at 25 °C and were analyzed in the general-purpose mode corresponding to the specialized software specifying the type of particle and dispersant used.

#### 3.3.2. Zeta Potential Analysis

The surface charge (ZP) of all formulations (aqueous and reconstituted freeze-dried PEG-cSLNs) was measured via laser Doppler electrophoresis in a Zetasizer Nano-Z (Malvern Instruments, Malvern, UK). The zeta potential values were obtained from the electrophoretic mobility (EM) of the nanoparticles under an electric field through electrodes that are connected to the cell containing the sample. Specific software was used to apply the Henry equation to convert the EM of the SLNs into ZP values. Measurements were carried out in triplicate and stabilized at 25 °C before measurement, and the results were expressed in millivolts (mV).

### 3.4. Effect of PEGylation on the Physicochemical Characteristics of CO-cSLNs

This analysis was designed to assess the impact of PEGylation on three variables (PSD, PdI, and ZP) in cSLNs with PEG. The statistical analysis was performed using the parametric unpaired Student’s t-test due to the fulfillment of the required conditions (normality test). Additionally, a comparative analysis of the three variables in each formulation was conducted to examine potential differences between the two manufacturing procedures employed. The parametric paired Student’s t-test was utilized when conditions permitted, while the non-parametric Wilcoxon test was used in cases where the conditions were not met.

### 3.5. Freeze-Drying of Nanoparticles

#### 3.5.1. Glass Transition Temperature (Tg)

Thermal behavior was conducted using a differential scanning calorimeter (DSC), model 821e (Mettler Toledo, Columbus, OH, USA) before lyophilizing the PEG-cSLNs to establish the appropriate parameters for the process. Samples were weighed out (approx. 30 mg) into aluminum pans (40 µL) and hermetically sealed with aluminum covers. Nitrogen liquid was used to cool down the samples, and an empty aluminum pan was used as a reference. The assay was performed by freezing the samples from 25 °C to −80 °C at 10 °C/min. The sample was kept at this temperature for 1 min and then heated to 25 °C at 10 °C/min. The analyses were conducted within a nitrogen atmosphere (50 mL/min).

#### 3.5.2. Freeze-Drying

The PEG-cSLNs were freeze-dried using a solution of trehalose (5%, *w*/*v*) as a cryoprotectant on a LyoLab C85 20 (Coolvacuum, Barcelona, Spain) pilot freeze-drying system. Glass vials were placed on the shelf of the freeze-dryer. Each vial was semi-stoppered and contained 4 mL (2 mL of PEG-cSLNs suspension and 2 mL of trehalose solution, 5%), representing each of the five different nanoparticle formulations. Briefly, conventional ramp freezing was carried out, in which the lyophilization cycle consisted of freezing the samples at −55 °C for 2 h. Then, the temperature was increased to −15 °C with a vacuum of 0.30 mbar for 20 h (primary drying), and secondary drying was performed at 25 °C for 20 h. After lyophilization, the PEG-cSLNs were evaluated for the presence of a homogeneous white cake and uniform appearance, indicating successful lyophilization without moisture. The lyophilized powder was reconstituted with 4 mL of Milli-Q water and filtered (43–48 μm) prior to performing the experiments.

#### 3.5.3. Comparison between the Different Conditions after Freeze-Drying

This analysis compared the three variables of each formulation between PEG-cSLNs suspended in aqueous medium and freeze-dried PEG-cSLNs. The analysis was performed using the paired Student’s t-test when the conditions were met, and the Wilcoxon test otherwise.

### 3.6. Morphological Analysis of the PEG-cSLNs

The surface and content homogeneity of all nanoparticle formulations (suspended and reconstituted freeze-dried PEG-cSLNs) were analyzed by means of transmission electron microscopy (TEM). Images were taken using a Tecnai Spirit microscope equipped with a LaB6 cathode (FEI Company, Hillsboro, OR, USA). Images were recorded at 120 kV using a 1376 × 1024-pixel CCD Megaview camera. The samples were adsorbed onto carbon-coated copper grids and negatively stained with a 4.0% methylcellulose solution.

### 3.7. Physical Stability Study

The stability study was carried out by determining the particle size, PdI, and zeta potential values on a Zetasizer Nano ZS90 (Malvern Panalytical Ltd., UK) and Nano-Z (Malvern Instruments, UK) for both suspended and reconstituted freeze-dried PEG-cSLNs. Suspended samples were stored at 4 °C and 25 °C for one month, while the lyophilized PEG-cSLNs were stored for one year at the same temperatures mentioned before. For the correlation analysis, Pearson or Spearman tests were performed depending on the normality of the samples.

### 3.8. Biological Characterization of PEG-cSLNs

#### 3.8.1. Cytotoxicity Assay

Cell cultures were established with human embryonic kidney 293 T (HEK293T) cells obtained from the American Type Culture Collection (ATCC, Manassas, VA, USA). The cells were grown and maintained in DMEM with low glucose supplemented with 10% *v*/*v* heat-inactivated FBS (Life Technologies Corp., Eugene, OR, USA), penicillin/streptomycin, 4 mM L-glutamine, and non-essential amino acids at 37 °C in a 95% air and 5% CO_2_ atmosphere as previously described [45].

For the viability/cytotoxicity assays, HEK293T cells were grown in 35 mm plates M6 (Falcon Enamelware, London, UK) to approximately 70–80% confluence. Two different volumes (10 μL and 20 μL) of each PEG-cSLN, in both suspended and freeze-dried states, were incubated with HEK293T cells in duplicate. Cellular cytotoxicity was assessed using flow cytometric analysis with the propidium iodide (PI) test. Briefly, HEK293T cells were seeded in 6-well plates (2.5 × 105 cells/well) and incubated in DMEM supplemented with 10% FBSi at 37 °C for 24 h. The culture medium was removed, the nanoparticles were added to DMEM without antibiotics or FBSi, and the cells were incubated again at 37 °C for 48 h. Flow cytofluorometric analyses were performed with the vital dye PI (40 μg/mL, EMD Millipore) using a FACSCalibur cytometer (BD Biosciences, San Jose, CA, USA). A minimum of 10,000 events were acquired by gating the forward and side scatters to exclude cell debris and analyzed in FL-3.

#### 3.8.2. SLNplexes Formation

SLNplexes were prepared by mixing the PEG-cSLNs (10 μL) with different amounts (500 ng, 750 ng, 1000 ng, and 2000 ng) of plasmid DNA (pDNA). The mixture was kept at room temperature (RT) for 40–45 min to allow the complexes to form. The analysis of SLNplex formation and loading efficiency of the PEG-cSLNs with pDNA was performed by examining the electrophoretic mobility of the samples across a 1.0% agarose gel. Gels were made with agarose D-1 at 0.8% in 1× Tris-acetate-EDTA (TAE) (40 mM Tris-acetate, 1 mM EDTA) containing 0.04 μL/mL RedSafe^®^ solution for nucleic acid visualization. An aliquot of each sample was mixed with 6X loading dye (NEB B7025S) comprising 2.5% Ficoll-400, 10 mM EDTA, and 3.3 mM Tris–HCl (pH 8.0−25 °C) and charged into the respective gel well. Electrophoresis was carried out at 80 V for 45 min. The fragments were visualized under UV light using a UV transilluminator GelDoc^®^ EZ Imager (Bio-Rad^®^, Hercules, CA, USA) system and photographed by Bio-Rad^®^ ImageLab 5.2.1 software.

### 3.9. Statistical Analysis

Three main analyses were carried out to establish whether there were significant differences between the different conditions to which the SLNs were exposed (PEGylation, protocols, nanoparticles suspended in an aqueous medium, and reconstituted freeze-dried nanoparticles). Two batches of each formulation were measured in triplicate to obtain the means and standard deviation of the particle size distribution (PSD), polydispersity index (PdI), and zeta potential (ZP).

All statistical tests and graphs were performed using IBM SPSS Statistics software, version 22 (SPSS, Chicago, IL, USA) and GraphPad Prism version 8.0.1 (GraphPad Software, San Diego, CA, USA). The *p* values are represented by asterisks (* *p* = 0.01–0.05, ** *p* = 0.001–0.01, and *** *p* < 0.001). The absence of an asterisk indicates that the change relative to the control is not statistically significant. The SLNs without PEG represented the control, whose denomination was CO due to cholesteryl oleate.

## 4. Conclusions

PEGylation and modifications in the manufacturing method affected the physicochemical characteristics of the CO-cSLNs. PEGylation significantly reduced particle size, and the lyophilization process influenced the physicochemical properties of PEG-cSLNs. However, successful reconstitution of PEG-cSLNs was achieved with a monodisperse population and positive surface charge, making them suitable for DNA binding and complex formation.

Furthermore, PEGylation notably improved the stability of suspended CO-cSLNs (more than one month). Most freeze-dried PEG formulations (4 of 5) showed stability for one year. The biological properties of CO-cSLNs remained unaltered by PEGylation, as cell viability remained high (over 90%). The consistent surface charge over time facilitated SLNplex formation, highlighting their potential as non-viral vectors.

Finally, these findings demonstrated the potential of these PEGylated nanoparticles for advancing transfection tests and conducting subsequent in vivo assays. The use of cost-effective PEGs could further enhance their commercial accessibility, making them promising candidates for future biomedical applications.

## Figures and Tables

**Figure 1 pharmaceuticals-16-01583-f001:**
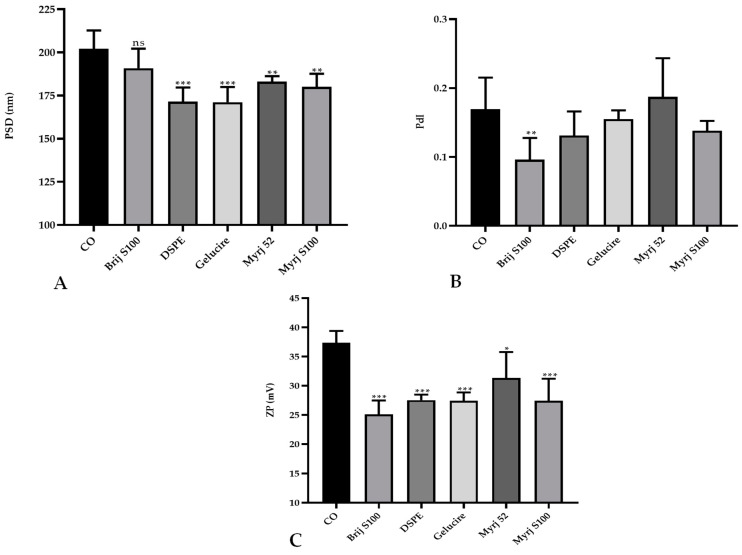
Comparative analysis of the (**A**) particle size distribution (PSD), (**B**) polydispersity index (PdI), and (**C**) zeta potential (ZP; C) between PEGylated and non-PEGylated nanoparticles. CO represents cholesteryl oleate, corresponding to the formulation without PEG. Statistical analysis was performed in triplicate. Bars without an asterisk (*) are not significant (ns), * *p* = 0.01–0.05, ** *p* = 0.001–0.01 and *** *p* < 0.001.

**Figure 2 pharmaceuticals-16-01583-f002:**
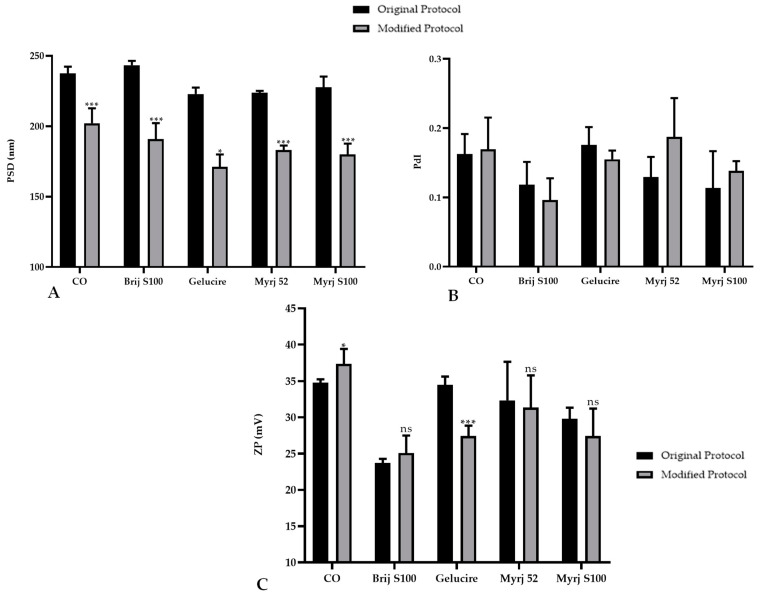
Comparative analysis of the (**A**) particle size distribution (PSD), (**B**) polydispersity index (PdI), and (**C**) zeta potential (ZP) of PEGylated and non-PEGylated nanoparticles with two different protocols. CO represents cholesteryl oleate, corresponding to the formulation without PEG. Statistical analysis was performed in triplicate. Bars without an asterisk (*) are not significant (ns), * *p* = 0.01–0.05 and *** *p* < 0.001.

**Figure 3 pharmaceuticals-16-01583-f003:**
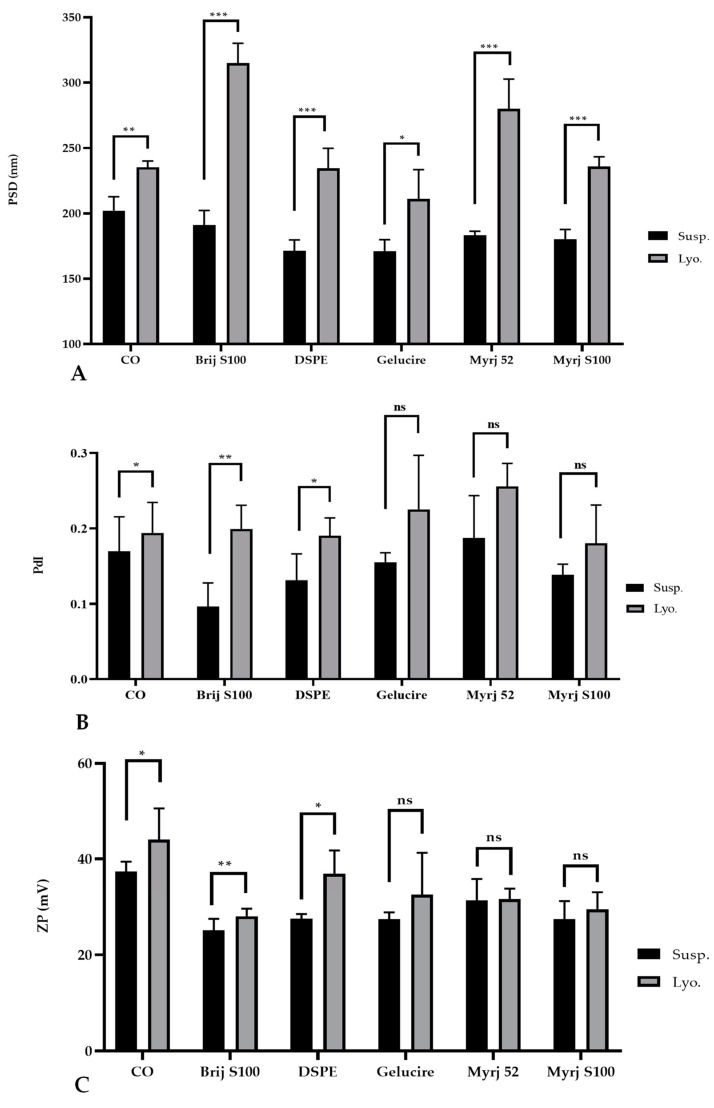
Comparative analysis of the (**A**) particle size distribution (PSD), (**B**) polydispersity index (PdI), and (**C**) zeta potential (Z) of suspended (Susp.) and freeze-dried (Lyo.) PEGylated and non-PEGylated nanoparticles. CO represents cholesteryl oleate, corresponding to the formulation without PEG. Statistical analysis was performed in triplicate. Not significant (ns), * *p* = 0.01–0.05, ** *p* = 0.001–0.01 and *** *p* < 0.001.

**Figure 4 pharmaceuticals-16-01583-f004:**
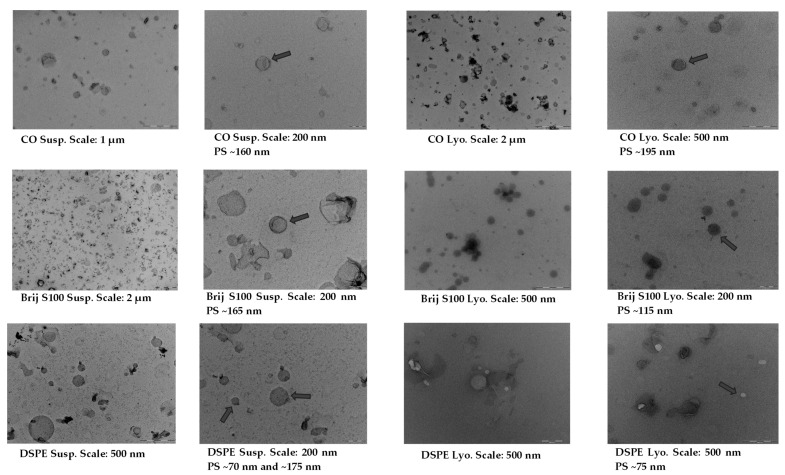
TEM images and morphological analysis of suspended and freeze-dried PEGylated and non-PEGylated nanoparticles. Abbreviations: Susp., nanoparticles suspended in an aqueous solution; Lyo., lyophilized nanoparticles; CO, cholesteryl oleate corresponding to the formulation without PEG. Gray arrows indicate particle size (PS).

**Figure 5 pharmaceuticals-16-01583-f005:**
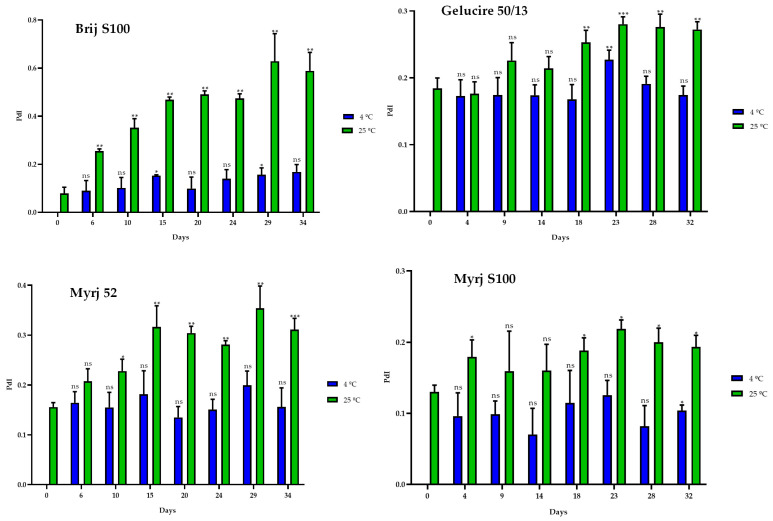
Polydispersity index (PdI) of the four PEGylated nanoparticles over a 30-day period at different temperatures (4 °C and 25 °C). Statistical analysis was performed in triplicate. Not significant (ns), * *p* = 0.01–0.05, ** *p* = 0.001–0.01 and *** *p* < 0.001.

**Figure 6 pharmaceuticals-16-01583-f006:**
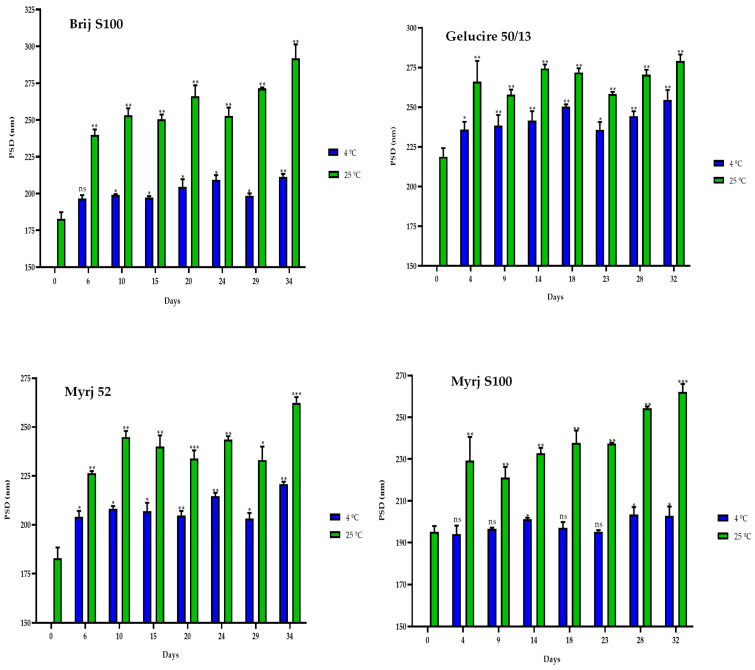
Particle size distribution (PSD) of the four PEGylated nanoparticles over a 30-day period at different temperatures (4 °C and 25 °C). Statistical analysis was performed in triplicate. Statistical analysis was performed in triplicate. Not significant (ns), * *p* = 0.01–0.05, ** *p* = 0.001–0.01 and *** *p* < 0.001.

**Figure 7 pharmaceuticals-16-01583-f007:**
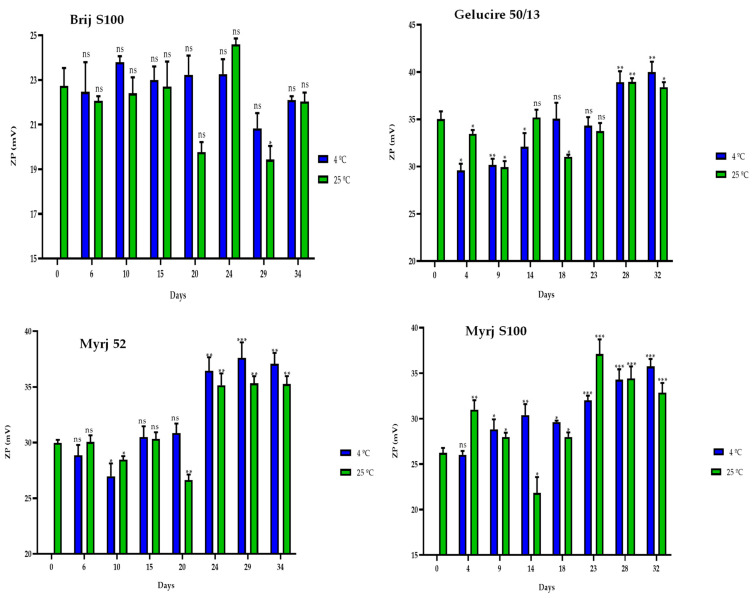
Zeta potential (ZP) of the four PEGylated nanoparticles over a 30-day period at different temperatures (4 °C and 25 °C). Statistical analysis was performed in triplicate. Not significant (ns), * *p* = 0.01–0.05, ** *p* = 0.001–0.01 and *** *p* < 0.001.

**Figure 8 pharmaceuticals-16-01583-f008:**
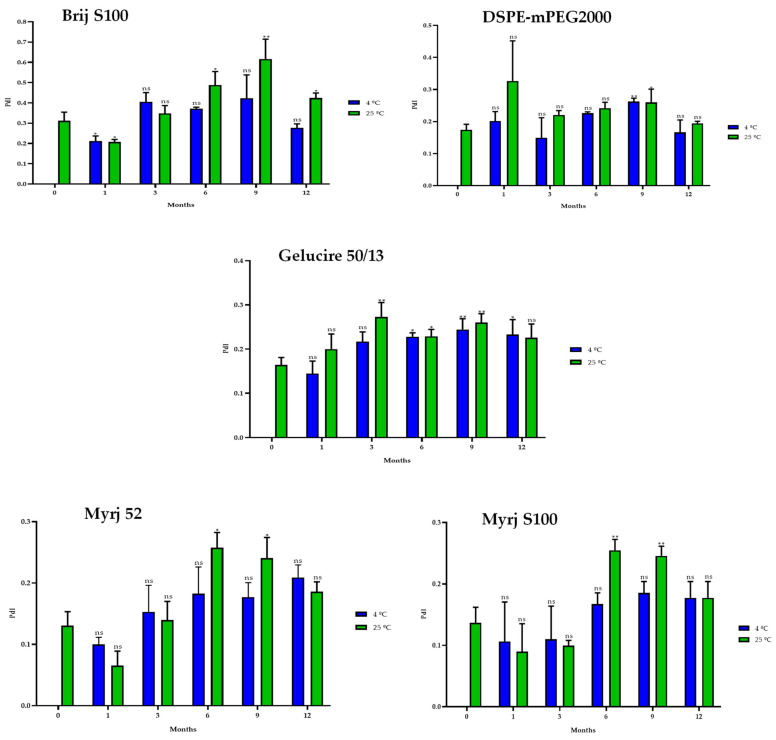
Polydispersity index (PdI) of the five PEGylated nanoparticles over one year at different temperatures (4 °C and 25 °C). Statistical analysis was performed in triplicate. Not significant (ns), * *p* = 0.01–0.05 and ** *p* = 0.001–0.01.

**Figure 9 pharmaceuticals-16-01583-f009:**
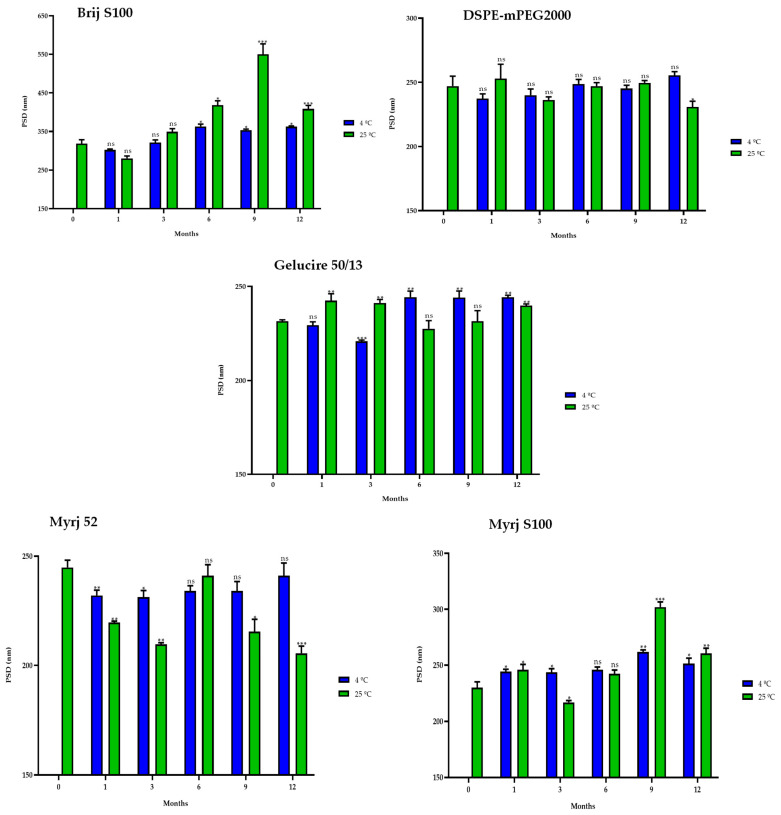
Particle size distribution (PSD) of the five PEGylated nanoparticles over one year at different temperatures (4 °C and 25 °C). Statistical analysis was performed in triplicate. Not significant (ns), * *p* = 0.01–0.05, ** *p* = 0.001–0.01 and *** *p* < 0.001.

**Figure 10 pharmaceuticals-16-01583-f010:**
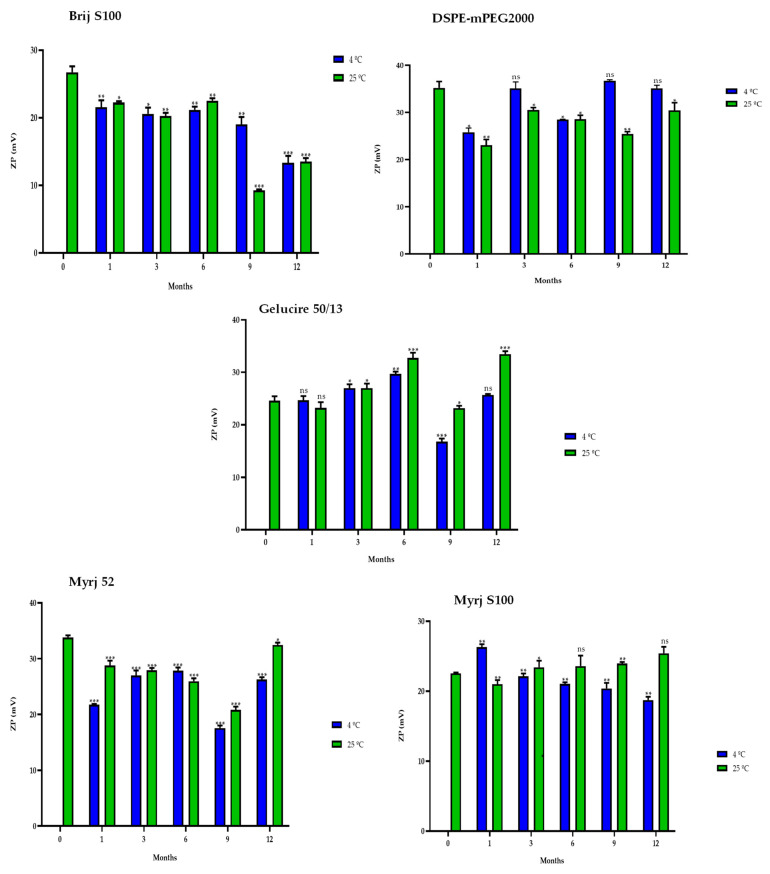
Zeta potential (ZP) of the five PEGylated nanoparticles over one year at different temperatures (4 °C and 25 °C). Statistical analysis was performed in triplicate. Not significant (ns), * *p* = 0.01–0.05, ** *p* = 0.001–0.01 and *** *p* < 0.001.

**Figure 11 pharmaceuticals-16-01583-f011:**
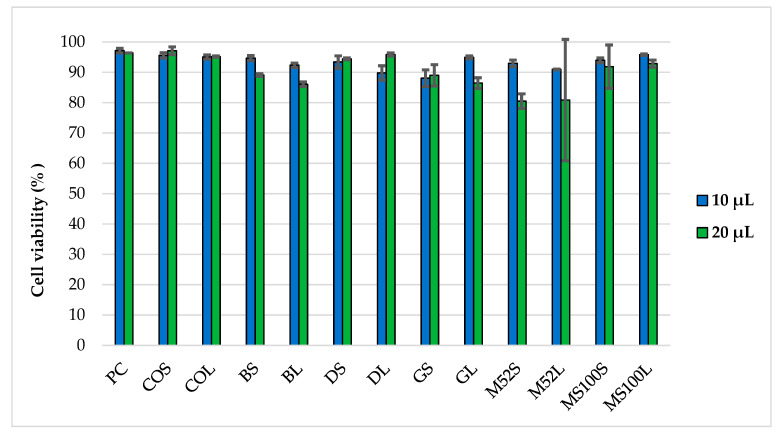
Cell viability of HEK293T cells after 48 h of incubation with two different volumes (10 μL and 20 μL) of PEG-cSLNs in suspended and lyophilized states using flow cytometry. Abbreviations: PC, positive control; COS, cholesteryl oleate suspended; COL, cholesteryl oleate lyophilized; BS, Brij S100 suspended; BL, Brij S100 lyophilized; DS, DSPE suspended; DL, DSPE lyophilized; GS, Gelucire suspended; GL, Gelucire lyophilized; M52S, Myrj 52 suspended; M52L, Myrj 52 lyophilized; MS100S, Myrj S100 suspended; MS100L, Myrj S100 lyophilized.

**Figure 12 pharmaceuticals-16-01583-f012:**
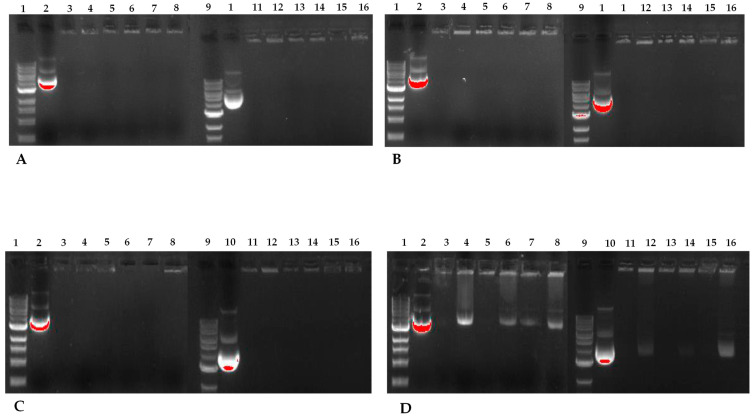
Agarose gel electrophoresis of the five different PEG-cSLNs–DNA complexes in suspended and lyophilized states with increasing amounts of plasmid DNA. Image (**A**) corresponds to 500 ng of DNA, image (**B**) to 750 ng, image (**C**) to 1000 ng, and image (**D**) to 2000 ng. Lanes 1 and 9: DNA ladder; lanes 2 and 10: MHC luciferase plasmid; lanes 3 and 4: cholesteryl oleate suspended and lyophilized; lanes 5 and 6: DSPE suspended and lyophilized; lanes 7 and 8: Myrj S100 suspended and lyophilized; lanes 11 and 12: Gelucire suspended and lyophilized; lanes 13 and 14: Myrj 52 suspended and lyophilized; lanes 15 and 16: Brij S100 suspended and lyophilized.

**Table 1 pharmaceuticals-16-01583-t001:** Comparison of the physicochemical characteristics of four different PEGs incorporated into the core of the cSLN matrix at varying quantities.

PEG ^1^	Quantity (mg)	PSD (nm)	PdI	ZP (mV)
Brij S100	50	348.8	0.252	23.4
	150	243.3	0.118	23.7
	250	216.6	0.156	20.8
	350	**166.2** ^2^	0.177	23.0
Gelucire	50	344.2	0.241	37.0
	150	**222.8**	0.176	34.5
	250	226.7	0.159	38.9
	350	264.6	0.185	33.1
Myrj 52	50	326.4	0.203	31.9
	150	**223.7**	0.129	32.3
	250	270.5	0.168	30.3
	350	246.8	0.153	28.9
Myrj S100	50	253.5	0.058	26.5
	150	227.6	0.114	29.8
	250	**222.8**	0.051	28.0
	350	296.6	0.086	29.5

^1^ Polyethylene glycol (PEG), particle size distribution (PSD), polydispersity index (PdI), and zeta potential (ZP). ^2^ Bold: the smallest particle size of each quantity evaluated.

**Table 2 pharmaceuticals-16-01583-t002:** Comparison of the physicochemical characteristics of four different PEGs incorporated into the core of the SLN matrix with two different protocols.

PEG (150 mg) ^1^	PSD (nm) ^2^	PdI	ZP (mV)	PSD (nm)	PdI	ZP (mV)
	Original Protocol			Modified Protocol		
Brij S100	243.3	0.118	23.7	190.90	0.10	25.10
DSPE	—	—	—	171.5	0.131	27.5
Gelucire	222.8	0.176	34.5	171.20	0.16	27.40
Myrj 52	223.7	0.129	32.3	183.10	0.19	31.40
Myrj S100	227.6	0.114	29.8	180.10	0.14	27.40
CO	237.4	0.163	34.8	202.10	0.17	37.40

^1^ DSPE-mPEG2000 is the most expensive PEG excipient (1 g = $500) used in this study. It was only used in the final essays. CO represents 300 mg of cholesteryl oleate, corresponding to the formulation without PEG. ^2^ Particle size distribution (PSD), polydispersity index (PdI), and zeta potential (ZP) of all the formulations manufactured.

**Table 3 pharmaceuticals-16-01583-t003:** Comparison of the physicochemical characteristics of nanoparticles in aqueous suspensions and lyophilized formulations with and without PEG.

PEG (150 mg) ^1^	PSD (nm) ^2^	PdI	ZP (mV)	PSD (nm)	PdI	ZP (mV)
	Suspended SLNs			Lyophilized SLNs		
Brij S100	190.9	0.096	25.1	315.1	0.199	28.0
DSPE	171.5	0.131	27.5	234.6	0.190	36.9
Gelucire	171.2	0.155	27.4	211.1	0.225	32.6
Myrj 52	183.1	0.187	31.4	280	0.256	31.6
Myrj S100	180.1	0.138	27.4	235.9	0.180	29.5
CO	202.1	0.169	37.4	235.3	0.194	44.1

^1^ CO represents 300 mg of cholesteryl oleate, corresponding to the formulation without PEG. ^2^ Particle size distribution (PSD), polydispersity index (PdI), and zeta potential (ZP) of all the formulations manufactured.

## Data Availability

The data presented supporting the conclusions established in this study are available within this article.

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
