# Peer review of "Comparative Analysis of the Physicochemical and Biological Characteristics of Freeze-Dried PEGylated Cationic Solid Lipid Nanoparticles"

_pharmaceuticals, 2023, doi:10.3390/ph16111583_

Round 1
Reviewer 1 Report
Comments and Suggestions for Authors
In this report, the authors presented the comparison between various PEGylated cSLNs. The work is interesting however in many places I have seen a lack of discussion after presenting the results. Therefore, I would recommend the proper addition of discussion along with results before publication. Here are some specific comments.
Title:
I would recommend shortening the title
Abstract:
Abbreviate PEG, DSPE, and Myrj S100 at their first appearance.
Introduction:
Written well and Informative.
Abbreviate RNA, DNA, COVID, and LNP at their first appearance
Results and Discussion:
In the first section 2.1. The results are presented however the discussion is missing. I would recommend adding a proper discussion on how various PEGs affect the particle size. What happens when adding different quantities of various PEGs to the formulation? The discussion must be supported by the literature. Focus on the formulation’s physicochemical properties.
Section 2.1.2: “Paliwal et al. (2014) reported that particle size could be reduced by increasing the duration (min) and stirring speed (rpm) during the manufacturing process.” Instead of writing this sentence write “Why increasing the speed and time can be responsible for decreasing the size of NPs”?
Line 152-153: Here the RPM stirring was increased from 20000 to 24000, this seems too much for any instrument. Mention which magnetic stirrer was used it should be homogenization speed. However, I would suggest checking the speed or instrument again that was used in the preparation.
Material and Methods
Explained well.
Reviewer 2 Report
Comments and Suggestions for Authors
Dear authors, Thank you for the opportunity to see your work. I do not find any critical errors in the analysis of the results and conclusions. However, I believe that the manuscript cannot be published in its current version. In the case of research describing the properties of new materials, it is key and critical to precisely characterize the methods of producing the new material!!!
Meanwhile, in your work the entire characterization looks like this: "The PEG-cSLNs were manufactured using an oil-in-water (o/w) emulsion technique based on the hot microemulsification method, as previously described [43], with some modifications to reduce particle size" - end of the description.
Please, describe here in details the method of PEG-cSLN generation. This is necessary because the reader must be able to repeat your research - or conduct other, similar, complementary research exactly in the same conditions.
Therefore, it is necessary to describe in detail the "hot microemulsification particle generation" method used, the modifications of this method and all the equipment used. As an example, in the mixing description it is necessary not only to specify the homogenization speed and its time (20000rpm and 10 minutes - as in 43 ref. ???), but also the device used for homogenization, the shape and size of the homogenizing tip and the size (volume) of the homogenized sample (i.e. the shape and volume of the vessel in which homogenization was carried out) . Similarly, all stages of particle production, equipment, parameters, etc., etc. should be described in detail. Only in this form will your work be suitable for publication.
Reviewer 3 Report
Comments and Suggestions for Authors
The manuscript entitled “Comparative Analysis of the Physicochemical Characteristics, Morphology, Stability, Cytotoxicity, and SLNplex Formation Capacity of Freeze-Dried PEGylated Cationic Solid Lipid Nanoparticles” delas with the development and detailed characterization of PEG-cSLNs, which is suitable for plex formation with plasmid DNA. The article is well-written, contains interesting information about SLN formulation, however I have some concerns, which should be clarified.
Please specify the description of DSC method! Did the authors applied any inert gas to avoid oxidation during the measurement? Further, did the authors investigate the thermal behaviour of components at the temperature of preparation (from 25 to +80°C)?
Please provide the DSC curves in the Supplementary Material!
Please modify the title of Section 3.7. to “Physical stability study” as only the colloidal parameters of cSLNs were tested.
The scale bar in the TEM images is not visible, please correct it!
Among the stability results there also seams to be some significant differences depending on storage temperature. Please also evaluate statistically these data!
Did the authors investigate, whether PEG-cSLNs can improve the stability of plasmid DNA in presence of DNase?
Round 2
Reviewer 2 Report
Comments and Suggestions for Authors
Dear authors
thanks for your improvements - and detailed presentation of method of PEG-cSLN generation
Reviewer 3 Report
Comments and Suggestions for Authors
The authors addressed all my concerns.